# A circuit motif in the zebrafish hindbrain for a two alternative behavioral choice to turn left or right

Minoru Koyama[1,2], Francesca Minale[3], Jennifer Shum[3], Nozomi Nishimura[3], Chris B Schaffer[3], Joseph R Fetcho[1*]

[1]Department of Neurobiology and Behavior, Cornell University, Ithaca, United States; [2]Howard Hughes Medical Institute, Janelia Research Campus, Ashburn, United States; [3]Meinig School of Biomedical Engineering, Cornell University, Ithaca, United States

**Abstract** Animals collect sensory information from the world and make adaptive choices about how to respond to it. Here, we reveal a network motif in the brain for one of the most fundamental behavioral choices made by bilaterally symmetric animals: whether to respond to a sensory stimulus by moving to the left or to the right. We define network connectivity in the hindbrain important for the lateralized escape behavior of zebrafish and then test the role of neurons by using laser ablations and behavioral studies. Key inhibitory neurons in the circuit lie in a column of morphologically similar cells that is one of a series of such columns that form a developmental and functional ground plan for building hindbrain networks. Repetition within the columns of the network motif we defined may therefore lie at the foundation of other lateralized behavioral choices.

*For correspondence: JRF49@ cornell.edu

**Competing interests:** The authors declare that no competing interests exist.

## Introduction

All animals have to collect sensory information from the environment and use it to produce adaptive behavioral choices. While theory, modeling, and experimentation has led to substantive ideas about circuit wiring that might effectively implement behavioral choice by vertebrate brains (*van Ravenzwaaij et al., 2012*; *Shadlen and Newsome, 2001*; *Drugowitsch et al., 2012*; *Doya and Shadlen, 2012*; *Bogacz et al., 2006*; *Faumont et al., 2012*; *Gaudry et al., 2013*; *Gaudry and Kristan, 2009*; *Shaw and Kristan, 1997*; *Song et al., 2015*; *Svoboda and Fetcho, 1996*), defining the cellular and synaptic organization of neurons involved in such choices has been more challenging. The solution to this problem might be more accessible in an evolutionarily old, conserved region of the brain, the hindbrain, which mediates what are some of the most fundamental behavioral choices made by vertebrate nervous systems: moving adaptively to avoid predators or to capture prey. The hindbrain is not only at the core of many sensory/motor responses, but also has a strikingly conserved ground plan of neuronal classes arranged early in life in columns ordered by structure and function (*Gray, 2013*; *Kinkhabwala et al., 2011*; *Koyama et al., 2011*). We set out to reveal a circuit motif for a simple, primitive behavioral choice mediated by hindbrain neurons within this vertebrate ground plan.

One of the most fundamental behavioral choices of all bilateral animals is whether to respond to the left or to the right by movements of the body, head, limbs and eyes. We studied a relatively simple, primitive, but critical lateralized behavior: the escape response of larval zebrafish, which involves a rapid turn away from a potential threat (*Faber et al., 1989*). This escape, like that in most aquatic vertebrates, begins with the firing of a single action potential in one of a bilateral pair of hindbrain

**eLife digest** Humans and other vertebrate animals constantly make choices about whether to respond to the left or to the right. Do they look left or right; turn left or right; reach left or right? In humans, the distinction between left and right is so fundamental that it has entered our collective thinking. Many societies define their political positions, for example, in terms of leaning to the left or to the right.

However, we know little about the wiring of the brain that accomplishes the task of making physical left-right choices. Koyama et al. therefore set out to identify the neural circuit responsible for the decision to turn either left or right. Zebrafish larvae were chosen as subjects because they execute rapid left or right turns to escape predators. Given that one wrong turn can result in the death of the zebrafish, a correct choice matters more than in most of the other decisions that animals make.

Experiments revealed that a process of competition between neurons on the left and right sides of the brain underlies this decision-making. Neurons on the right collect evidence that an attack is coming from the right, and drive turns to the left, away from the threat. These neurons also attempt to silence competing neurons on the left that act to produce turns to the right. By weighing up the evidence from left and right sides, the circuit as a whole comes to a decision about the best direction in which to turn.

The region of the brain that controls the left versus right escape response in zebrafish is present in all vertebrates. Moreover, it appears to have a similar structure across species, consisting of repeating columns of neurons. This raises the possibility that other left-right choices in fish and other animals occur in a similar way – a principle that can be tested in future work.

neurons, called Mauthner cells, whose axons cross in the brain and extend into the spinal cord. Activation of a Mauthner cell on one side initiates a turn away from an attacking predator that is the shortest latency and fastest motor response in vertebrates. Whether an escape is to the left versus the right is determined by which of the two cells fire, leading others to propose that this could be a simple model of decision making (*Korn and Faber, 2005*). A correct behavioral choice here is vital-one wrong turn in a lifetime can lead to death.

Earlier work showed that auditory inputs entering one side of the brain excite both the Mauthner cell and inhibitory neurons that were thought to be important for the laterality of the escape response (*Koyama et al., 2011*; *Faber et al., 1989*, *1978*; *Zottoli and Faber, 1980*; *Takahashi et al., 2002*); see diagram in *Figure 1A*. The inhibitory neurons lie at the bottom of a column of morphologically similar neurons within the hindbrain (*Figure 1B1–4*) that is one of a series of columns forming the ground plan from which the many circuits in vertebrate hindbrains arise (*Kinkhabwala et al., 2011*; *Koyama et al., 2011*). The repetitive structure of the hindbrain suggested that if we could determine the wiring and test the functional roles of neurons important for mediating the left/right choice in the Mauthner network, we would not only reveal a network motif with a role in a behavioral choice, but also one that is likely repeated within the columnar patterning to mediate other lateralized sensory motor responses of vertebrates.

## Results

### Connectivity of the neurons

We first determined the pattern and strength of the connections of the inhibitory neurons thought to play a role in the laterality of the escape because they are driven by auditory input and connect to the Mauthner cells. To reveal how they distributed their inhibition to the two Mauthner cells, we performed 19 triple patch recordings from individual feedforward inhibitory neurons and the two Mauthner cells, followed by high-quality dye filling of the three cells in 10 of the experiments. The neurons were targeted based on their location at the bottom of a lateral column of glycinergic neurons and the knowledge that at least some of the population straddled the lateral dendrite of the Mauthner cell (*Kinkhabwala et al., 2011*; *Koyama et al., 2011*). They could thus be targeted in a

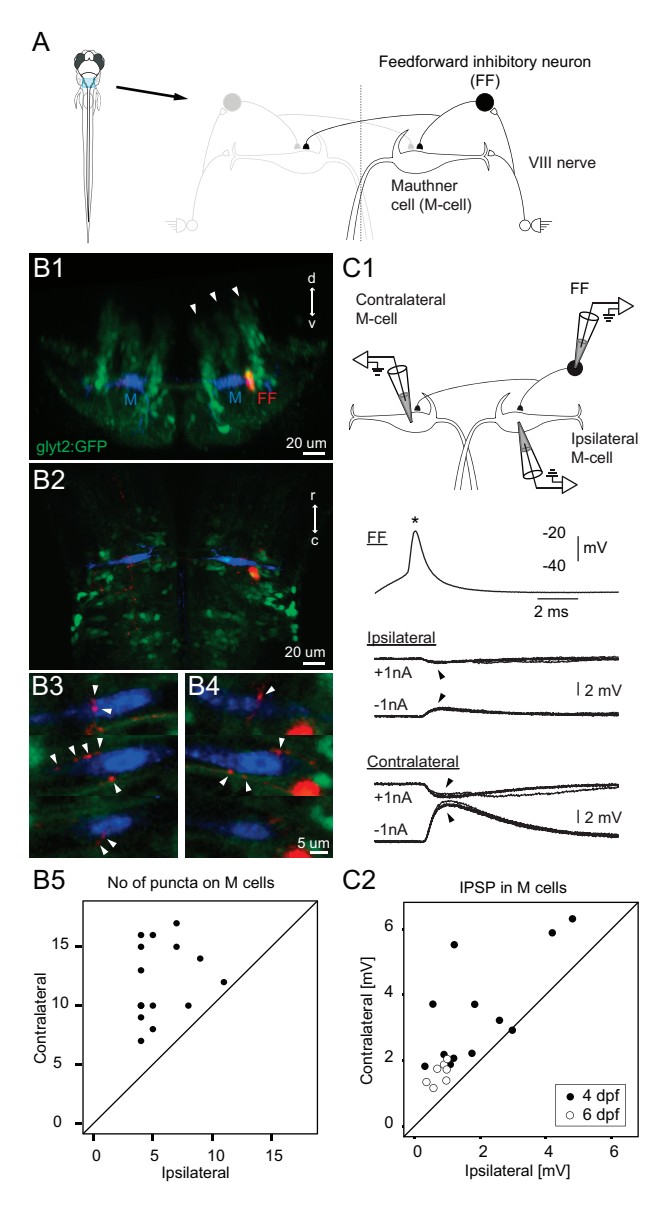

**Figure 1.** Connectivity of inhibitory neurons implicated in the laterality of the escape response. (**A**) The blue region marked on the drawing of the fish contains the two Mauthner cells and neuronal candidates for their control, which may determine whether the animal escapes to the left or right. Right side shows some known contacts in this network, with sensory inputs from the eighth nerve exciting the ipsilateral M-cell as well as inhibitory interneurons that inhibit both M-cells. (**B1**) A cross section through the hindbrain in the region of the Mauthner cell from a transgenic line with glycinergic neurons labeled with GFP (green). The two Mauthner cells (blue) and a filled feedforward (FF) inhibitory interneuron (red) were labeled via patch pipettes. The FF neuron lies at the bottom of the most lateral of three columns (arrowheads) of glycinergic neurons, near the lateral dendrite of the M-cell. (**B2**) A horizontal view of the region shows that red processes of the FF neuron are located in the vicinity of both the ipsilateral and contralateral M-cells. Optical sections through the region of the contralateral (**B3**) and ipsilateral M-cells (**B4**) show swellings (arrowheads) of the processes from the inhibitory FF cell adjacent to both M-cells. (**B5**) A plot of the number of boutons adjacent to ipsilateral versus contralateral M-cells from 14 fish (4 days old) in which individual FF cells and both M-cells were labeled. In every case, the number of boutons apposed to the contralateral cell exceeded the number apposed to the ipsilateral one, with a highly significant difference between the two sides ($p<0.0001$). (**C1**) Triple patch recordings from an FF neuron and the two M-cells show that firing the FF cell (asterisk) by current injection produces larger IPSPs in the contralateral M-cell than the ipsilateral one (arrowheads). IPSPs from four sweeps are shown at potentials above and below resting potential by

*Figure 1 continued on next page*

*Figure 1 continued*

the injection of plus or minus 1 nA of current into the M-cells. This electrophysiology is from the neurons whose morphology is shown in **B**. (**C2**) A plot of the amplitude of the IPSPs (for negative 1 nA injection) in the ipsilateral versus contralateral M-cell from 19 triple patch experiments like the one in **C1**. Seven of these were from 6 days old *nacre* fish and 12 from 4 days old *relaxed* fish. In 18 of 19 experiments, the IPSPs were larger in the contralateral M-cell, with a highly significant difference between the two sides (p<0.001).

line with GFP labeled glycinergic neurons by recording from GFP positive neurons located in that column of glycinergic cells and lying adjacent to the M-cell lateral dendrite, which was visible in DIC optics. Individual inhibitory neurons had commissural axons that branched on both sides of the body and had putative synaptic boutons apposed to both Mauthner cells as in *Figure 1B1–4*. In 14 cases where we filled the three neurons (10 with prior physiology and 4 only labeled anatomically to minimize cellular damage) and reconstructed them in 3D from confocal optical sections, every feedforward neuron had more boutons contacting the contralateral M-cell than the ipsilateral one, with a highly significant difference between the two sides (*Figure 1B5*; p<0.0001).

This clear morphological asymmetry was associated with a functional one revealed by firing the inhibitory neurons while recording the postsynaptic responses in both M-cells. *Figure 1C1* shows an example of one experiment in which the feedforward inhibitory neuron produced a larger IPSP in the contralateral M-cell than the ipsilateral one (morphology of this set is in *Figure 1B1–4*). This experiment was repeated at two different ages (4 and 6 days) in two lines of fish paralyzed either by bungarotoxin (in *nacre* line with reduced pigmentation), or by a mutation blocking calcium release in muscle (*relaxed* line; see Materials and Methods for details). The main result was the same independent of age, genetic line, and approach to paralysis. In 18 of 19 triple patch experiments, the strength of the contralateral IPSP was greater than the ipsilateral one and the overall difference was significant (mean contra/ipsi ratio was 2.44, range 0.97–6.61; p<0.001).

We conclude that individual inhibitory neurons are driven by ipsilateral sensory input and inhibit both M-cells, but at different strengths. The weaker inhibition of the ipsilateral M-cell, along with its direct excitation by excitatory sensory afferents might be expected to make that M-cell more likely to fire to an ipsilateral stimulus than the contralateral M-cell, resulting in an escape bend away from the stimulus source.

A sensory stimulus, however, will often activate sensory inputs on both sides of the body to differing extents, so the typical natural situation would be one in which the left and right populations of inhibitory neurons compete at the level of the two M-cells to influence which reaches threshold first. The problem of escaping a threat requires turning away from the side that receives the strongest sensory stimulus (typically the side of the attack) over a broad range of stimulus strengths above the minimum that signals a potential predatory attack. If the inhibitory neurons only competed at the level of the M-cells, asymmetric, but very large stimuli on the two sides might lead to massive inhibition of both M-cells, possibly delaying or blocking an escape.

The intuition that strong bilateral inputs might pose a problem with the known connectivity was examined more formally in a model incorporating our data from the inhibitory neurons (*Figure 2 A1–3* and *Figure 2—figure supplement 1*: *Table 1*). Here we focus on the versions of the model most closely tied to the experimental evidence, although all of the variations tested and their implications are presented in *Figure 2—figure supplement 1*. We initially modeled a circuit containing inhibitory neurons driven by sensory inputs, but with connections only to the ipsilateral M-cell. As expected, because the two M-cells are controlled independently, this network led to non-adaptive bilateral M-cell responses even when there were large differences in the input strengths on opposite sides of the body (*Figure 2B1*). Adding commissural inhibitory connections to the contralateral M-cell that were stronger than those to the ipsilateral one, in the proportions revealed by our data, allowed for some unilateral M-cell responses to strongly asymmetric bilateral inputs (*Figure 2B2*). The model performance was still flawed, however, because the strong crossed inhibition of the contralateral M-cell led to a broad range of strengths of bilateral sensory input over which neither M-cell fired, in line with our prior intuition that the strong commissural inhibition might not facilitate rapid, adaptive responses away from the side receiving the strongest input when inputs to both sides are substantial (*Figure 2B2*).

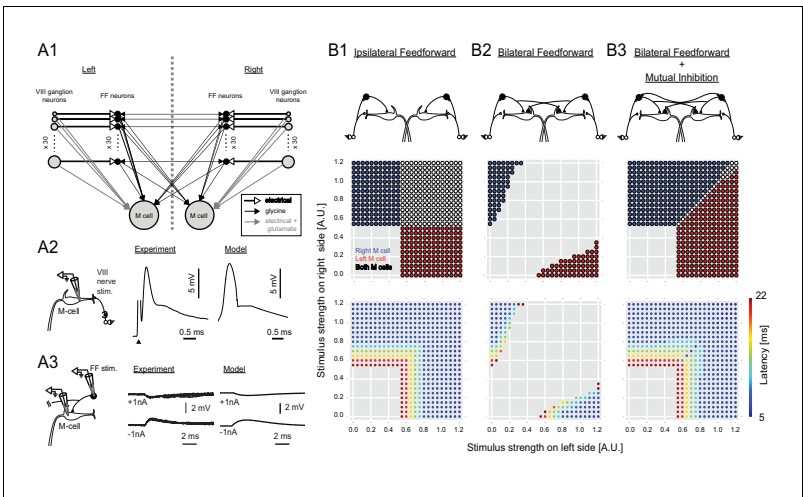

**Figure 2.** Output of a computational model of the studied neurons in the Mauthner network. (**2A1**) Connectivity implemented in the model. Gray arrows indicate mixed electrical and excitatory glutamatergic connections. White arrows indicate pure electrical connections. Black arrows indicate inhibitory glycinergic connections. (**2A2**) Mauthner response to eighth nerve stimulation. The eighth nerve on one side was stimulated (left panel) and the EPSP in the lateral dendrite of Mauthner cell was recorded experimentally (middle panel). The EPSP in the model was simulated by synchronous activation of eighth ganglion neurons (right panel). (**2A3**) Inhibitory input from an FF neuron to a Mauthner cell. A single actual FF neuron is activated by current injection to fire a single spike while monitoring IPSPs in the ipsilateral Mauthner cell (left panel). Four traces are shown at potentials above and below reversal potential by injection of plus or minus 1 nA of current into the M-cells (middle panel). The modeled IPSP in the ipsilateral Mauthner cell during injection of plus or minus 1 nA of current (right panel). (**2B1**) Output of the computational model of Mauthner circuit with only ipsilateral feedforward inhibitory connections (upper panel). Various combinations of left (x-axis) and right stimuli (y-axis) are presented to the model circuit and each stimulus condition is color-coded based on the activation of Mauthner cells (blue: only right Mauthner cell fires, red: only left Mauthner cell fires, white: both Mauthner cells fire) (middle panel) and also based on the latency of the activation (lower panel). (**2B2**) Output as in **B1**, but with the addition of asymmetric bilateral feedforward inhibitory connections (upper panel). Middle and lower panels were formatted as in **B1**. (**2B3**) Output as in **B2**, but with the addition of putative reciprocal inhibitory connections between the FF neurons (upper panel).

The following figure supplement is available for figure 2:

**Figure supplement 1.** Outcomes of all of the circuit configurations modeled.

One solution to this problem is for the inhibitory neurons to reciprocally inhibit one another, which might serve to reduce the overall level of inhibition during bilateral inputs and provide inhibition in proportion to the difference between the strengths of sensory inputs on the two sides of the body (*Mysore and Knudsen, 2012*). Our modeling shows that such a reciprocal connection could solve the problem of a lack of escapes to strong bilateral inputs (*Figure 2B3*). Such reciprocal connections between the inhibitory neurons were unknown, so we tested the prediction of their existence by using pairwise patch recording followed by intracellular labeling.

We recorded from 17 bilateral pairs of feedforward inhibitory neurons (10 in *relaxed* fish at 4 dpf, 7 in *nacre* fish at 6 dpf) and filled them with dye to confirm their identity. 14 of the 17 pairs were connected. Of these 14, 10 pairs were connected in just one direction (left cell inhibiting right, or right inhibiting left), as in the neurons shown in *Figure 3A–B*. The other 4 reciprocally inhibited each other at the single cell level as in *Figure 3C–D*. The connections were blocked by strychnine, consistent with the glycinergic phenotype of the feedforward neurons (*Figure 3C–D*). The distribution of patterns of connectivity is summarized in *Figure 3E1* for the two groups of fish. The amplitudes of the PSPs measured at rest ranged from 0.34 to 12.08 mV and averaged 3.85 mV (*Figure 3E2*), with no statistical difference in the distributions of strength in connections from right to left versus from left to right. In connected pairs, the processes of the presynaptic neuron were apposed to the postsynaptic cell (*Figure 3B1–2*; cell filled in the physiology experiment in 3A), as expected for a

**Table 1.** Experimentally measured properties and settings used in the Modeling. Top: Experimentally derived basic properties of Mauthner and Feedforward glycinergic neurons at 4 dpf. (Mauthner: n=24, FF: n=28). Rm, input resistance; Erest, resting membrane potential; ECl, reversal potential of IPSP; Espike, spiking threshold; Tau, membrane time constant. Corrected for liquid junction potential. Bottom: Parameters used for conductance-based models of the Mauthner circuit. Rm, input resistance; Erest, resting membrane potential; ECl, reversal potential of IPSP; Espike, spiking threshold; Tau, time constant; El, leak reversal potential; EK, reversal potential of potassium; ENa, reversal potential of sodium; gl, leak conductance per area; gNa, sodium conductance per area; gK, potassium conductance per area.

1. Measured parameters

|  | Mauthner | FF |
|---|---|---|
| Rm [Mohm] | 10.3 ± 0.9 | 411.7 ± 41.16 |
| Erest [mV] | −78.5 ± 0.9 | −77.3 ± 0.7 |
| ECl [mV] | −75.4 ± 0.94 | NA |
| Espike [mV] | −61.3 ± 1.1 | −60.9 ± 1.0 |
| Tau [ms] | 22.6 ± 7.4 | 9.9 ± 1.2 |

2. Model parameters

| LIF model parameters | Mauthner | FF |
|---|---|---|
| Rm [Mohm] | 10 | 400 |
| Erest [mV] | −79 | −77 |
| Ecl [mV] | −75 | −75 |
| Espike [mV] | −61 | −61 |
| Tau [ms] | 23 | 10 |
| No of Cells/side | 1 | 30 |

| HH model parameters | Auditory |
|---|---|
| El [mV] | −79 |
| EK [mV] | −90 |
| ENa [mV] | 50 |
| gl [msiemns/cm2] | 0.05 |
| gNa [msiemens/cm2] | 100 |
| gK[msiemens/cm2] | 200 |
| No of Cells/side | 30 |
| Surface area [um2] | 2000–20000 |

| Connectivity Parameters | Synapse type | Conductance [nS] | Delay [ms] | Connectivity pattern |
|---|---|---|---|---|
| FF -> ipsilateral M | Glycine | 25 | 0.3 | all to one |
| FF-> contralteral M | Glycine | 2.5 × FF->ipsiM | 0.3 | all to one |
| Auditory-> M | Gap | 15 | NA | all to one |
| Auditory-> M | Glutamate | 12 | 0.7 | all to one |
| Auditory->FF | Gap | 10 | NA | one to one |
| FF -> contra FF | Glycine | 12 | 0.3 | all to all |

| Synapse time constants | Glutamate |  | Glycine |
|---|---|---|---|
| Tau [ms] | 2 |  | 2 |

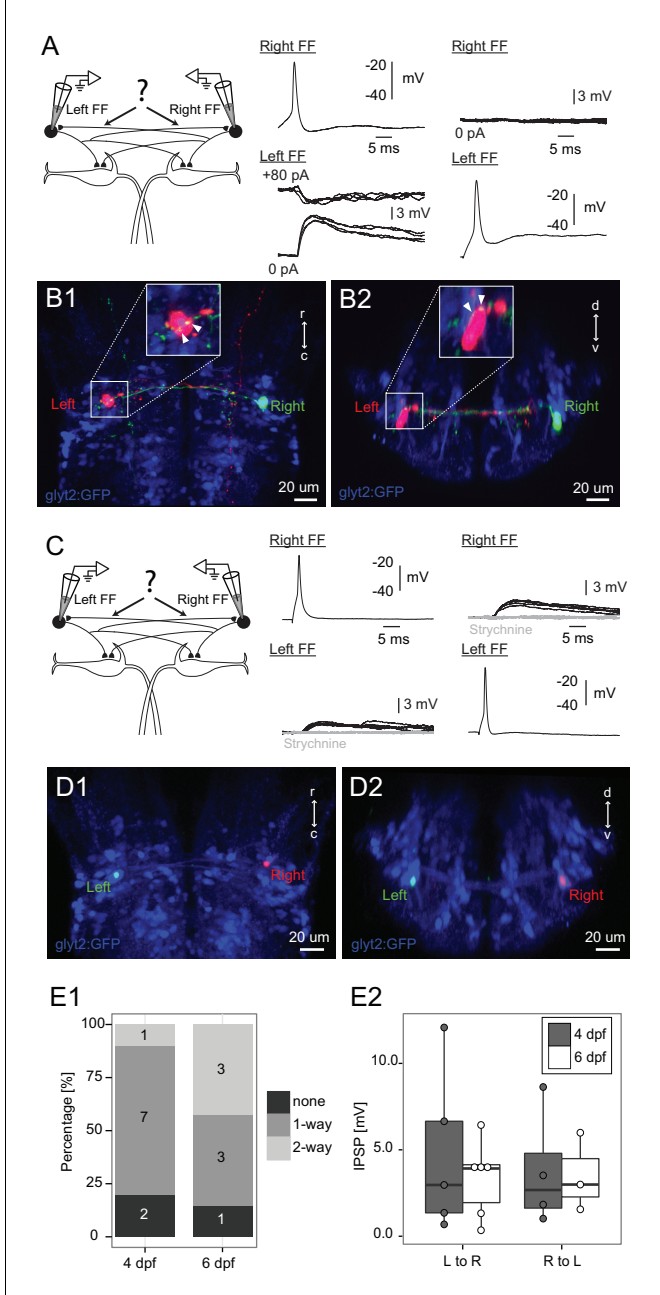

**Figure 3.** Connections between FF inhibitory neurons on the two sides. (**A**) Patching of pairs of FF neurons on opposite sides of the brain. In this case, firing the right cell (Right FF) led to an IPSP in the FF on the left side (Left FF) (middle panel), but the left cell did not produce a response in the right one (right panel). **B1** and **B2** show horizontal and cross sections respectively, of the neurons recorded in **A** after filling them with dye. Green processes from the right neuron give rise to swellings apposed to the left neuron (arrowheads on insets in **B1,B2**, which show the soma at higher magnification), consistent with the physiological connection. (**C**) Paired patch recordings of two reciprocally connected FF neurons. Firing the right neuron led to a depolarizing response in the left cell and vice versa (dark traces). Both responses were blocked by strychnine (1 µM, gray traces), consistent with the PSP being a glycinergic inhibition with a reversal potential just above resting potential in our recording conditions. (**D1–2**) The locations of the neurons recorded in **C** in confocal images of the dye filled neurons after recordings. (**E1**) Patching of 17 bilateral pairs of FF neurons from either 4 day (*relaxed*) or 6 day (*nacre*) old fish revealed that most were connected to each other in one or both directions. Numbers in the histograms indicate the number of pairs with connections in neither direction (none), one direction (1-way), or reciprocal connections (2-way). (**E2**) Box plot of the strengths of the connections from left to right and right to left (only measured in cases

*Figure 3 continued on next page*

*Figure 3 continued*
where there was a connection in a given direction). The connection strengths were not significantly different in the two directions.

monosynaptic connection. The neurons, like those reported in *Figure 1*, also had processes adjacent to the contralateral M-cell, indicating that individual neurons might inhibit both that M-cell and contralateral feedforward inhibitory cells.

## Testing of behavioral predictions

The pattern of connections in the network revealed by our experiments, along with the modeling, led to two predictions about the behavioral role of the inhibitory neurons in the initiation of the escape turn. The first, suggested by others before us, is that these neurons may contribute to determining the time at which an M-cell fires, which is when the decision to escape is made (*Faber et al., 1989*, *1978*). Sensory inputs directly excite the ipsilateral M- cell, but also inhibit it by exciting the feedforward inhibitory neurons that project to that M-cell. This leads to the expectation that the timing of the escape is set by a balance between direct excitation and feedforward inhibition, which would determine when an M-cell reaches threshold to a unilateral stimulus and thus when a choice to escape is made. The model incorporating known connection strengths also shows this is possible, as lowering the level of inhibition shortens the response latency to a unilateral stimulus because the excitation dominates and drives the cell to threshold sooner (All Ablated, Half Ablated, *Figure 4A*).

The second prediction is that the inhibitory neurons are playing a central role in determining which of the two M-cells fires, and thus the laterality (initial left or right bend) of the escape behavior. This contribution of the inhibitory neurons to laterality would be mediated both by their stronger inhibition of the contralateral M-cell and by the inhibition of their contralateral inhibitory counterparts. The model leads to the prediction that removing inhibitory neurons on one side will lead on average to more escapes initiated by the Mauthner cell on the opposite side when the balance of sensory inputs on the two sides is varied (*Figure 4B*). Thus, the experimentally derived network structure and the modeling suggest roles for the inhibitory neurons both in determining **when** a choice is made and **what** choice is made. We set out to test these two predictions by assessing behavioral performance following laser ablation of the feedforward inhibitory neurons.

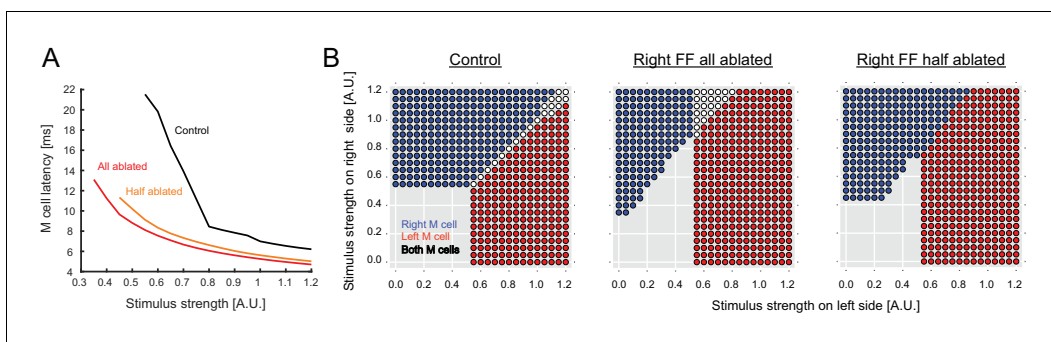

**Figure 4.** Output predicted by the model after ablation of FF neurons. (**A**) Modeled latency of the Mauthner cell spike as a function of the simulated stimulus strength in control (black) and in cases where the entire pool of FF-neurons was removed from the model on the stimulated side (red) or half the pool was removed (orange). The response latency becomes shorter after removal of FF neurons in the model. (**B**) Biased left-right decision after one-sided FF-removal in the model. A series of left and right stimuli is presented to the control model (left panel) and the FF-removed models (right panels). The stimulus conditions that lead to the activation of one of the Mauthner cells are color-coded based on the activation pattern (blue: only right Mauthner cell fires, red: only left Mauthner cell fires, white: both Mauthner cells fire). The likelihood of left Mauthner activation increases after the removal of right FF neurons.

The first prediction was that removal of the inhibitory neurons on one side would reduce the sensory driven inhibition of the ipsilateral M-cell, while leaving the excitation unchanged, thus shifting the balance to excitation and leading to both a quicker depolarization to threshold and a shorter latency escape response to a stimulus on that side. To test this, we used a high-energy, long wavelength femtosecond laser to ablate inhibitory neurons on one side in a transgenic line in which glycinergic neurons were GFP labeled and the M-cell was labeled by single cell electroporation with Texas red dextran. As a control for the specificity of the lesioning and possible damage to the M-cell, we targeted the laser to other glycinergic neurons near the Mauthner cell, but medial to the ones in the escape network. We also directly targeted the M-cell's lateral dendrite to either sever the dendrite from the cell body, or kill the M-cell.

The approximate laser powers for killing the M-cells or severing their lateral dendrite were established by a systematic exploration of the effects of pulses at different energy levels (*Figure 5*). These data led us to use levels of 70–90 nanojoules (nJ) of energy to kill or cut the lateral dendrite of the M-cell because these outcomes occurred with high frequency at those levels. We used lower levels of 50-55 nJ for ablation of the smaller glycinergic neurons, and still lower energy pulses in control, non-ablation laser exposure (0-20 nJ for glycinergic neurons. The high energy pulsed laser we used ablates by production of a plasma rather than thermal damage. As a consequence, collateral damage is minimized and it produces very specific lesions of single cell precision in XY and Z at the power levels we used, as shown for an inhibitory neuron in *Figure 6*. The effectiveness of all of the perturbations and the lack of obvious damage to adjacent neurons was confirmed in confocal or two

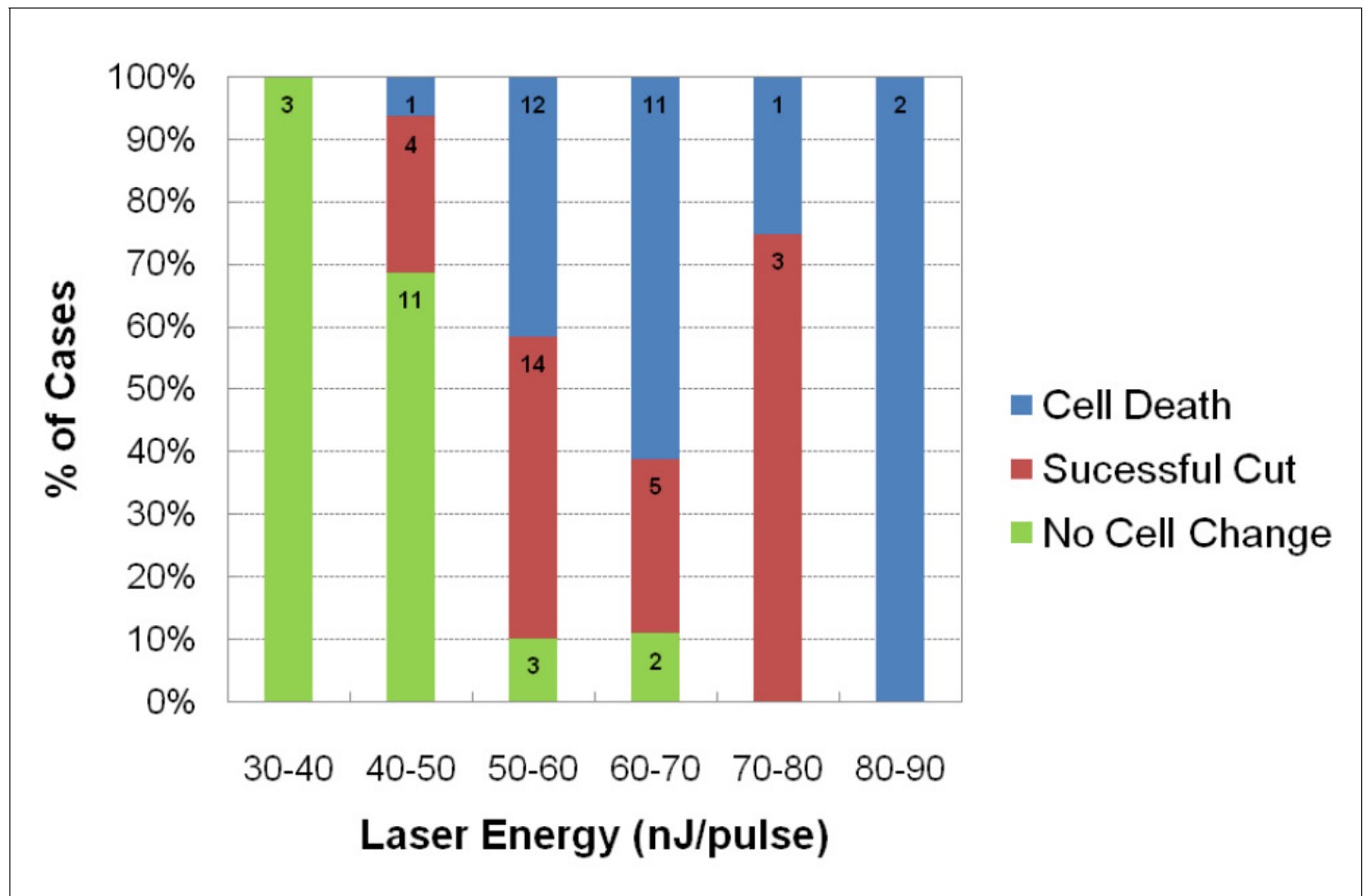

**Figure 5.** Assessment of impact of laser intensity on the Mauthner cell. Plot shows the results of targeting the lateral dendrite of 72 Mauthner cells with laser pulses of different intensities, showing the percentage of cases in which the cell died, the lateral dendrite was cut off with the neuron surviving, or no obvious change in the cell. Numbers on bars indicate the number of fish in each category.

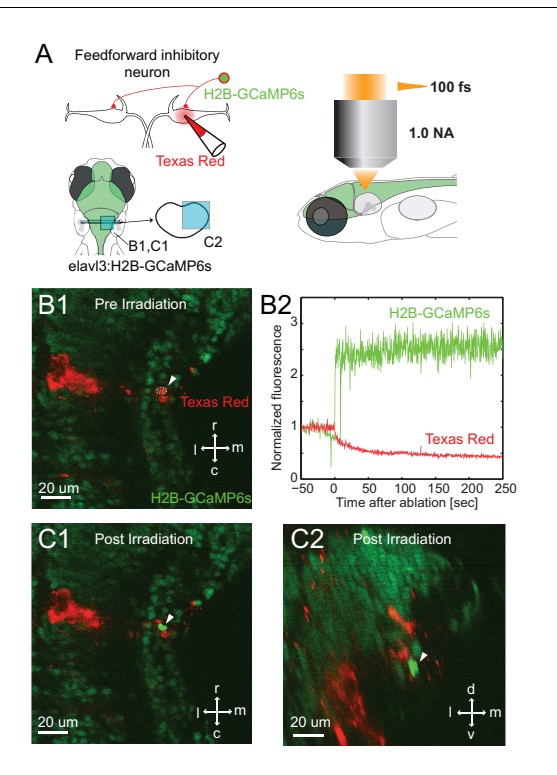

**Figure 6.** Specificity of laser targeting of inhibitory neurons. (**A**) FF neurons were backfilled with Texas red in a transgenic line with neurons labeled with nuclear targeted GCaMP6s. (**B1**) Image prior to laser targeting showing the cell to be targeted marked by the white arrow head. (**B2**) GCaMP6s fluorescence from the targeted cell following laser illumination for ablation shows a massive increase in fluorescence, also visible in images of the targeted neuron in horizontal (**C1**) and cross section (**C2**) images taken after illumination. Note the bright green signal is evident only in the targeted cell and not in surrounding neurons next to and above it, showing the specificity of the targeting.

photon 3D reconstructions from the live fish at the end of all of the experiments, as illustrated in the examples in *Figure 7*.

The experiments involved six comparison groups, all with unilateral targeting of the laser: control sham lesions directed at the feedforward neurons, but at laser energy below levels producing damage; ablation of feedforward glycinergic neurons; ablation of glycinergic neurons not implicated in escapes, that are located adjacent to the M-cell soma, but in the most medial glycinergic column (see *Figure 1B1,2*) about 40 µm medial to the feedforward cell type; cutting off the lateral dendrite of the M-cell which lies adjacent to the somata of the inhibitory neurons; and ablation of the M-cell.

We targeted 4 to 8 inhibitory neurons for ablation per animal. The number of cells that lost fluorescence in the post-ablation stack averaged 5.2 (s.d.=1.3). We do not have a firm way of establishing the overall size of the feedforward inhibitory population, but we estimated it based on backfills from the Mauthner cell on one side. We found most filled cells on the contralateral side were located in the ventral half of the lateral glycinergic stripe (*Koyama et al., 2011*). Because backfilling is very likely to miss some cells, to estimate the overall size of the inhibitory population we counted the number of glycinergic cells in the lateral stripe in rhombomere 4 located ventral to the most dorsal backfilled neurons, whether the cells in that region were backfilled or not. This led to an average of 31.3 cells (se=0.7, n=8 fish). We suspect this is an overestimate (possibly a large one) of the number that actually connect to the M-cell because the assumption here is that all the glycinergic cells in that region connect to the M-cell. If we take this value as an upper limit of the size of the population, we estimate that we will have removed minimally 13–25% of the population. The neurons we targeted with the laser were reliably identifiable based on their position relative to the lateral dendrite,

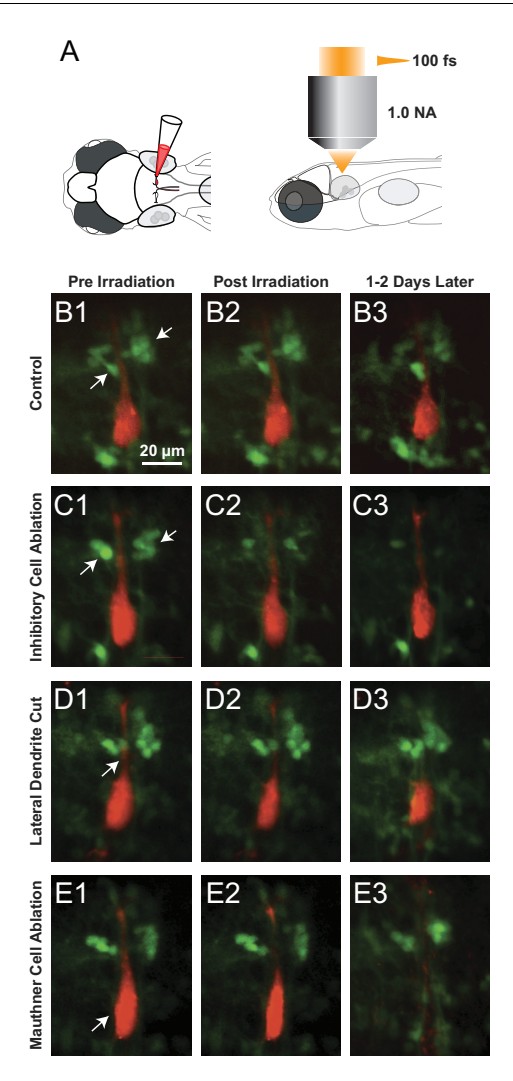

**Figure 7.** Examples of Laser ablations. (**A**) Diagram showing the approach to ablations, with initial filling of the Mauthner cell with red dye by single cell electroporation in a line with GFP labeled glycinergic neurons, followed by imaging and ablations from above with a femtosecond laser. (**B1–B3**) Control fish exposed to laser power below threshold for lesions shown pre irradiation (**B1**), immediately post irradiation (**B2**) and between 1 and 2 days later (**B3**). Glycinergic neurons are green, with arrows pointing to the cluster of feedforward cells on either side of the red Mauthner cell. (**C1–C3**) As in **B**, but with laser ablation directed at the visible feedforward neurons (arrows in **C1**), most of which were removed (**C3**), while preserving the adjacent red dendrite of the M-cell. (**D1–D3**) Laser targeting of the lateral dendrite of the Mauthner cell (arrow) sometimes severs the lateral dendrite, while leaving the adjacent inhibitory neurons and the soma of the M-cell intact (**D3**). (**E1–E3**) Laser targeting of the lateral dendrite that led to death of the M-cell (soma marked by arrowhead in **E1** is absent in **E3**), but preservation of adjacent inhibitory neurons.

and we were confident based on our many pairwise recordings that neurons in this location were members of the feedforward population.

The day after the ablations, we tested the escape latency to a highly directional stimulus – a pressure-ejected pulse of water directed at either the lesioned or unlesioned sides of the caudal body/tail of the fish. This gave precise temporal control of the stimulus as well as a predominantly unilateral stimulus. A tail stimulus was chosen because it is known to reliably engage the M-cell (*Liu and Fetcho, 1999*; *O'Malley et al., 1996*). In addition, we showed that this stimulus, though tail directed, excites auditory afferents based upon recordings from the eighth nerve during such a

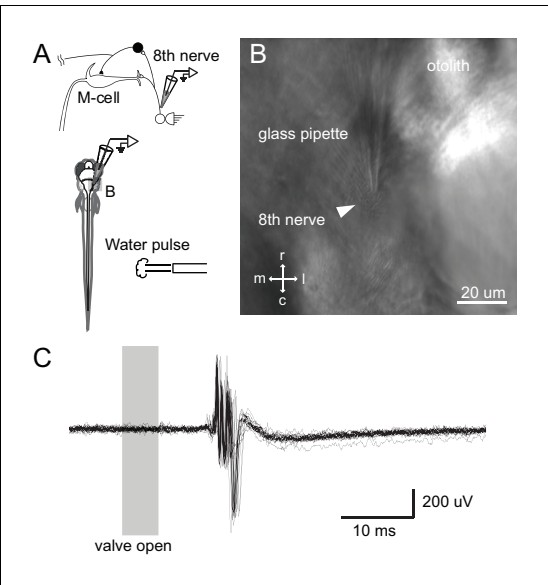

**Figure 8.** Tail directed water pulse leads to activation of axons in the eighth cranial nerve. (**A**) Diagram of the experiment in which extracellular recordings from the eighth cranial nerve in a paralyzed larvae were performed while applying a pulse of water directed at the tail. (**B**) Image of the glass recording electrode on the eighth nerve. (**C**) The water pulse led to extracellularly recorded activity in fibers of the eighth nerve, showing that a water pulse like that we used on freely swimming fish can produce an auditory response, even though tail directed.

stimulus (*Figure 8*). The stimulus is, however, very likely multimodal, as we expect it to excite somatosensory and lateral line inputs, along with auditory ones.

The results of these experiments are shown in *Figure 9*. In control animals, with inhibitory neurons targeted with an attenuated laser, the median latency for the initial turns following a stimulus was 11 ms (mean=16.4 s.e. = 2.2, n=38) on one side and 12 ms on the other (mean=13.6 ms s.e.=1.4, n=31), with no significant difference between the two sides. Control lesions of medial inhibitory neurons near the M-cell, but not implicated in escape, led to median latencies of 10 ms with no difference between the two sides (mean=11.3 ms s.e.= 0.8, n=73; mean=11.5 ms s.e.=0.7, n=86). Killing the M-cell produced a large and significant increase in latency to a median of 42 ms (mean= 36.8 s. e.=4.3, n=17; significance values in *Figure 9* caption) to a stimulus on the lesioned side and a significant difference in latency between the lesioned and unlesioned sides of the fish, consistent with earlier work (*Liu and Fetcho, 1999*). Severing the lateral dendrite of the M-cell led to a trend toward increased latencies on the lesioned side to a stimulus on that side (median = 13.0 ms, mean=16.0, s. e.=1.6, n=51), but the lesioned and unlesioned sides were not significantly different.

Most importantly, however, removing the feedforward inhibitory neurons caused an effect opposite to that seen by directly targeting the adjacent M-cell and one not seen after removal of other nearby glycinergic cells or sham ablations. Ablation of the feedforward neurons (Lateral glycine ablation in Fig. 9B) significantly shortened the latency of the response to a stimulus on the lesioned side (median=6.0 ms, mean=8.3 ms, s.e.=1.7, n=26), so that it was about half of the latency of the escape response of control fish and significantly different from the latency to stimuli on the unlesioned side (median=10.0 ms, mean=11.9 ms, s.e.=1.9, n =27). This difference in the effect of removing the feedforward inhibitory cells versus damaging the immediately adjacent Mauthner cell or other inhibitory neurons near it supports the specificity of the lesions and argues against the observations arising from a general nonspecific effect of damage in the region.

Lesions of inhibitory neurons thus 'improved"' the fish's response time by shortening the latency to respond. This is consistent with the prediction that the inhibitory neurons help to determine when the decision to escape is normally made by elevating the sensory stimulus required for the M-cell to reach threshold. A relatively high response threshold may be important to avoid production of escapes to weak, non-threatening stimuli.

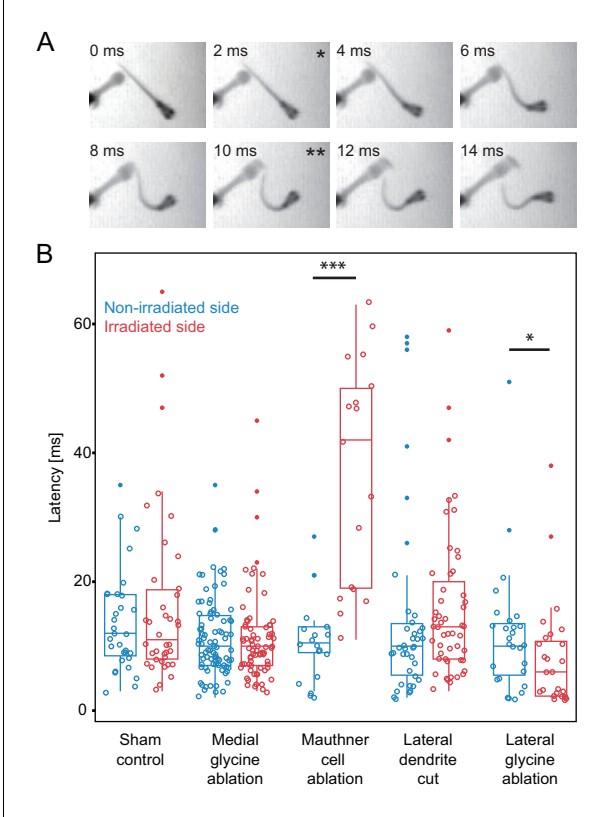

**Figure 9.** Escape latency of control and lesioned fish in response to a unilaterally directed squirt of water. (**A**) Example trial showing the escape elicited by a squirt of colored water from a needle on the left side of the frame. The bend begins in the second frame (*) and the fish performs a rapid escape bend away from the stimulus in subsequent frames, with the peak initial bend at 10 ms (**). (**B**.) All of the data for the latency of escape response from lesioned (red) and unlesioned (blue) sides in controls and in the different ablation conditions. Box and whisker plots represent median as well as first and third quartiles. Outlier points are solid. Asterisks mark significant differences between responses on intact and lesioned sides (*p<0.05, ***p<0.001, corrected for multiple comparisons). Sham ablations and ablation of medial glycinergic cells near the M-cell, but not implicated in escapes, did not affect the latency. Killing the Mauthner cell led to a significant increase in latency. Cutting dendrite of the M-cell also led to a trend toward a longer latency, although it was not significantly different from the intact side. Ablation of the lateral FF glycinergic neurons led to a significant reduction in the latency to respond.

A unilaterally directed stimulus was useful for revealing how the balance of excitation and inhibition affects the time to the response, but many normal stimuli will activate sensory inputs bilaterally to varying extents, with the left/right direction of the escape determined by which of the two M-cells fires first. The circuit we defined suggests that each side competes with the other to control the choice of escape direction by inhibiting the M-cell and the feedforward inhibitory neurons on the opposite side. We infer, and our modeling supports, the possibility that the inhibition might act to directly suppress the contralateral M-cell and also indirectly promote activation of the ipsilateral one by blocking the inhibition coming from the other side. The circuit configuration, synaptic strengths, and modeling lead to the prediction that if escapes were elicited by stimuli over a range of directions and strengths, then the removal of FF inhibitory cells on one side would result in more escapes initiated by the opposite, intact side.

We tested this prediction by eliciting escapes in a situation in which the fish was freely swimming in a dish on the surface of a vibration plate that produced a sudden, brief vertical displacement of 9 g acceleration (*Figure 10A,B*; see materials and methods). Because the location and orientation of the fish varied in the dish from trial to trial we expected that the balance of sensory input to the two

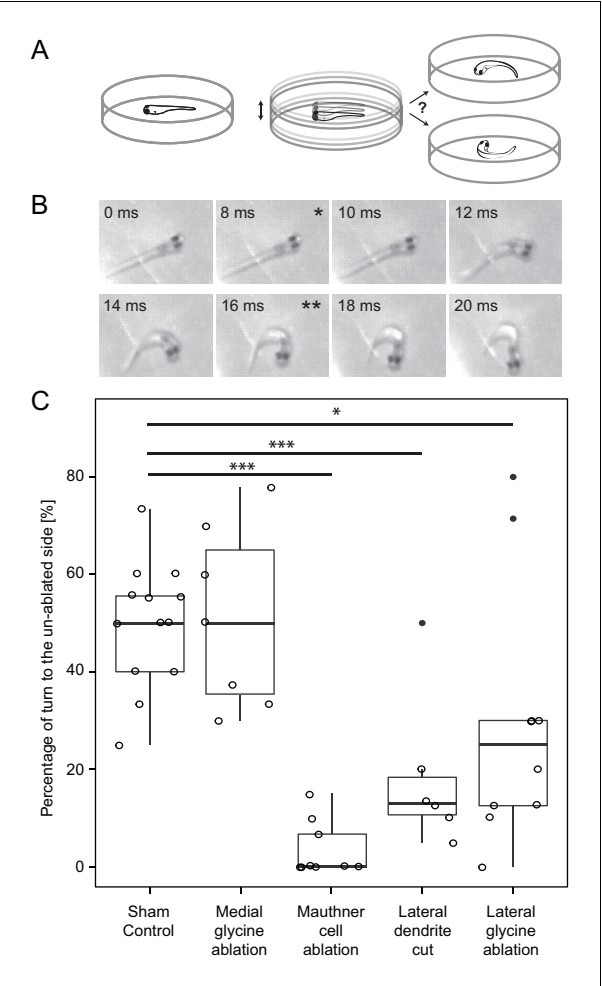

**Figure 10.** Response laterality of control and lesioned fish in response to an omnidirectional stimulus. (**A**) Diagram of the experiment in which a dish containing a freely swimming fish is vibrated and the left/right direction of the initial escape turn is monitored. (**B**) An example of the high speed video recordings of a trial in which the escape response occurred to the fish's right side. Asterisks mark the beginning (*) and end (**) of the initial bend. (**C**) All of the data for response laterality for the control and lesion conditions, showing the percentage of responses initiated by the ablated side, which led to a turn away from that side and toward the intact side. Every lesion condition, except the ablation of medial glycinergic cells not implicated in escape, was significantly different from sham controls, with the magnitude of the change differing depending upon the type of lesion (*p<0.05, ***p<0.001, corrected for multiple comparisons).

sides (and thus the perceived location of its source) would also vary from trial to trial leading, over a series of trials, to an equal number of escape bends to the left and to the right in an intact fish. *Figure 10C* shows that unlesioned control fish did on average produce half of their responses to the left and half to the right.

We decided a priori to focus on laterality and not on response latency in these experiments because the dish displacement stimulus did not offer as narrow a time estimate as the squirting for when the fish sensed the stimulus. Nonetheless, the large majority of the responses were short latency, rapid C-bends consistent with those produced by involvement of the M-cell. For example, in 134 responses from control and feedforward lesioned fish, the mean latency was 10.1 ms (s.e.=0.82; only 6 were longer than 20 ms) measured from the onset of the dish displacement. Responses with the kinematics of an escape (*Liu and Fetcho, 1999*) of all latencies were included in the analysis.

We asked how different laser perturbations changed the ratio of left/right responses to the omni-directional stimuli. The outcome of the removal of a control set of medial glycinergic neurons

adjacent to the M-cell was like that of sham ablations, with no resulting imbalance between the two sides. Killing the M-cell on one side led to the strongest effect with a median of zero percent of short latency escapes produced by the lesioned side (mean=3.5, s.e.=1.9, n=9), consistent with short latency turns being driven largely by the intact M-cell when the other M-cell is eliminated. Cutting off the lateral dendrite of the M-cell also led to a clear bias in the laterality of responses, with more short latency escape bends initiated by the side opposite to the lesion. The median percentage of responses initiated by the lesioned side was only 12.9 percent (mean=18.5, s.e.=6.6 n=6). This result might be expected because substantial excitatory input onto the cut dendrite is removed by the cut, thereby reducing the ability of the lesioned cell to respond to the stimulus, thus biasing responses to the intact side.

The most important result, however, in the context of the role of the FF inhibitory neurons in behavioral choice is the effect of their ablation on response laterality. After unilateral removal of a portion of the FF inhibitory population, we found a statistically significant drop in the fraction of escapes produced by the lesioned side (Lateral glycine ablation, *Figure 10 C*; median=25, mean=29.6, s.e.=8.3, n=10 p<0.05). The majority (70–75%) of escapes were produced by the intact side, as predicted. In summary, the changes in both behavioral paradigms were those predicted based upon the patterns and strengths of the neuronal connectivity we revealed, supporting the conclusion that we have defined a network motif important for a simple two alternative behavioral choice.

## Discussion

Even simple nervous systems have to select alternative motor responses based on sensory evidence. In bilateral animals like vertebrates, the laterality of a response is one of the most fundamental and primitive behavioral choices as it is critical for avoiding predators and navigating through the environment, however big a brain the animal might have. Our work was directed toward revealing a network motif that contributes to this basic behavioral choice in vertebrate nervous systems.

One advantage of Mauthner system we studied is that these two cells are the locus of a left right decision. If one of the Mauthner cells fires, the fish generates a strong bend (*Nissanov et al., 1990*). The firing of one of the two cells thus represents a choice of laterality, potentially offering insight into simple networks for behavioral choice – here considered in the broadest sense as collecting sensory evidence and using it to select an adaptive motor response from possible alternatives (*Korn and Faber, 2005*).

In escapes, like most behaviors, correct behavioral choices require producing the appropriate behavior at the right time. The connectivity motif of the excitatory, and inhibitory interneurons defined here (*Figure 11A*) contributes to solving both of these problems through its influence on both the laterality of an escape bend and its timing. The key aspects of the motif are excitatory inputs conveying sensory evidence (here, auditory inputs) for the alternatives to the neurons driving the output of the choice (here, the right or left M-cell). The choice depends on a competition between the two alternatives, as sensory evidence for one choice inhibits the other through inhibitory neurons that suppress the alternative outcome (the contralateral M-cell). Reciprocal connections between those inhibitory neurons likely assure that the response is fast and lateralized even for strong bilateral inputs by providing inhibition to choice neurons (M-cells) that scales with the difference in the level of sensory input on the two sides rather than the magnitude of the sensory drive. Finally, the timing of the response is determined by a balance between excitation and inhibition at the choice point (the M-cell), including a feedforward inhibition that likely prevents escape responses to weak inputs. The circuit motif we describe has a pattern of connections similar to network structures that theoretical work shows can allow for optimal choices between two alternative responses based on sensory evidence, also supporting its likely importance for behavioral choice (*Bogacz et al., 2006*; *Mysore and Knudsen, 2012*).

The roles of the motif in the timing and lateralization of the escapes are supported by lesion experiments that produced outcomes predicted by the connectivity pattern we revealed, including changes in the laterality of the response and the ability to shorten an already very short latency response. Extensive controls support the specificity of the perturbations. Our simple model incorporating the known connections and strengths could produce adaptive responses away from the side of the strongest input over a very broad range of bilateral sensory strengths, showing the possibility

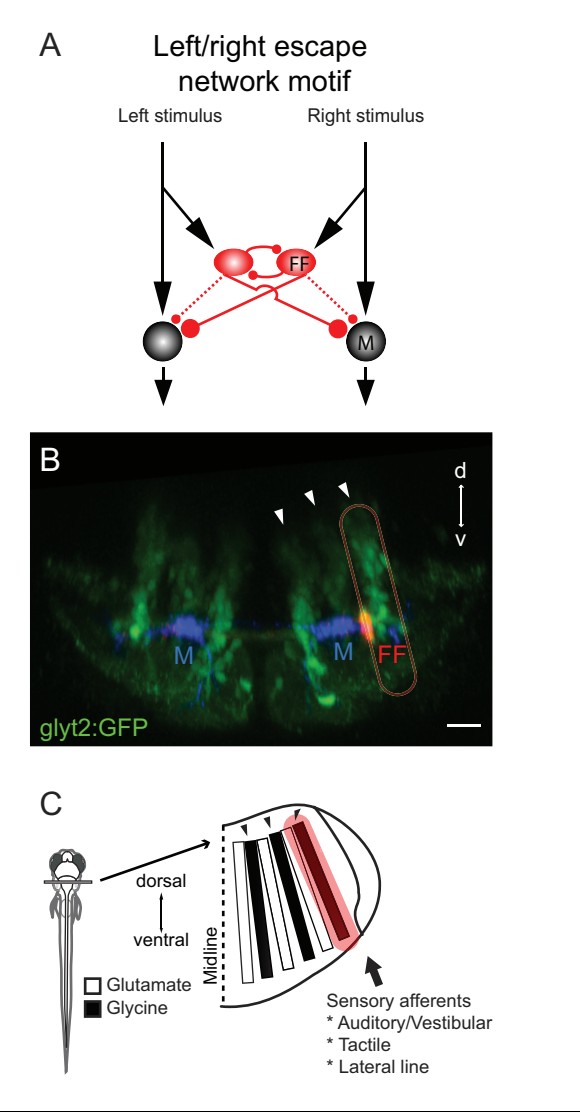

**Figure 11.** Summary of the circuit and its relationship to hindbrain structure. (**A**) Diagram of the circuit motif for the left right choice revealed and tested behaviorally in this paper. Relative synaptic strengths are indicated by contact size. Dotted line shows ipsilateral output of the FF. M: Mauthner cell, FF(red): Feedfoward inhibitory neuron. (**B**) Image of the disposition of the neurons in the hindbrain. The inhibitory neurons (red, FF) lie at the bottom the most lateral of three columns of glycinergic inhibitory neurons (green) marked by arrows. This lateral column contains commissural inhibitory neurons with outputs on both sides of the hindbrain, while more medial columns contain neurons with contralateral (middle column) or ipsilateral outputs (medial column). Scale bar=20 μm. (**C**) The column containing the FF neurons is one of a series of interleaved glutamatergic and glycinergic columns in the hindbrain. The variety of sensory inputs to the lateral glycinergic column and the shared morphology of the neurons in it raise the possibility that other neurons in the column might also contribute to the laterality of other behavioral responses to sensory inputs by implementing the same circuit motif.

that such a network, even with a small number of cell types, could account for the behavior – a proof of principle at least. Importantly, removal of neurons in the model led to results consistent with lesion experiments. In combination with the outcomes of perturbations in the model that were not possible experimentally, the model points to the importance of the pattern of connections for producing escape responses away from the side of the strongest stimulus at relatively short latency (but not to weak stimuli) over a range of bilateral stimulus strengths. Together, the experimental

evidence and the modeling support a key role for the circuit motif we revealed, shown in *Figure 11A*, in determining the timing and laterality of a behavioral choice.

Here, the timing and laterality of the escape bend depend on inhibitory as well as excitatory inputs, in contrast to recent work suggesting that the side of the initiation of swimming in tadpoles is determined by an imbalance in excitation reaching reticulospinal neurons on opposite sides (*Buhl et al., 2015*). That work detected no inhibition, even though the possibility for it was left open. A purely excitatory mechanism might falter if a precise choice of the laterality of the response is required, as in the escape behavior. In the initiation of swimming, the choice of the side on which the propagating wave starts may not be as critical as the direction of a turn away from a predator.

Importantly, the feedforward inhibitory neurons are not the only neurons that might contribute to the features of the escape response in different contexts. The Mauthner cell network also has excitatory interneurons that can influence its firing (*Lacoste et al., 2015*), and there are very likely other unknown inputs as well that could also have an effect on which cell fires when. In addition, while the timing and laterality of the response is determined by when one of the two M-cell fires, the magnitude of the response depends on patterns of activation of other neurons in the hindbrain and spinal cord (*Liu and Fetcho, 1999*; *O'Malley et al., 1996*; *Kohashi and Oda, 2008*).

Given the many lateralized behaviors (eye movements, head movements, limb movements) produced by animals and their variations in magnitude, we might expect that a circuit motif underlying lateralized responses in the brain would be repeated to control different lateralized behaviors. While on the face of it, the escape network seems very specialized, we now know that it arises in an orderly way from a columnar ground plan in the hindbrain that increasing evidence suggests is shared by vertebrates broadly (*Figure 11A–C*) (*Gray, 2013*; *Kinkhabwala et al., 2011*; *Koyama et al., 2011*). The feedforward inhibitory neurons that we have shown are key to determining the threshold and directionality of escapes are a small subset of one of the hindbrain columns that contains other bilaterally projecting glycinergic inhibitory neurons. This raises the possibility that the other morphologically similar neurons in the column might also receive sensory inputs and have similar patterns of connectivity, but with different output neurons than the M-cell, to implement left right choices in other networks that move the eyes, head, body and limbs.

Prior studies of both the spinal cord and hindbrain in zebrafish suggest how repetition of such circuit motifs might be organized (*Kinkhabwala et al., 2011*; *McLean et al., 2007*; *McLean and Fetcho, 2009*; *McLean et al., 2008*). Early in life, hindbrain neurons are ordered by age and function, which roughly map onto their dorsoventral position in larval zebrafish. In the hindbrain columns prior to neuronal migration (much of which occurs after larvae have hatched and already have the ability to swim, escape and feed to survive), older, more ventral neurons like the M-cell, the ventral feedforward inhibitory neurons, and ventral descending excitatory neurons in a more medial column (positive for the Chx10/Alx transcription factor) control powerful movements such as escape and fast swimming (*Kinkhabwala et al., 2011*; *Koyama et al., 2011*). Increasingly younger, more dorsal neurons are recruited for successively slower, weaker swimming responses. Lateralized motor responses can also vary widely in strength as well as latency to respond; for example, escape turns produced by old neurons are at the most powerful end of a range of turning movements that also include much slower and longer latency turns (*Budick and O'Malley, 2000*; *McElligott and O'malley, 2005*; *Burgess and Granato, 2007*). We therefore predict that younger inhibitory neurons located more dorsally in the column containing the escape feedforward cells might share patterns of connectivity defined here, but will collect sensory information over increasingly longer time scales to drive appropriately lateralized behaviors that occur at longer latencies and reduced strengths than the escape. Longer latency responses might be produced by neurons with longer integration times (there is already evidence for a gradient of integration times in zebrafish oculomotor circuits that maps onto location in the hindbrain [*Miri et al., 2011*]) or by the accumulation of evidence via a gradual recruitment within a larger population of neurons than is engaged in the very fast escape behavior. If so, then repeated network motifs in the hindbrain may contain sensory motor circuits that accumulate sensory information over different time scales to drive behavioral choices to turn left or right over a range of different latencies, speeds and magnitudes.

# Materials and methods

## Animal husbandry

All experiments were performed on 4 to 7 day post-fertilization (dpf) zebrafish obtained from a laboratory stock of transgenic and mutant adults. All procedures conform to the US National Institute of Health guidelines regarding animal experimentation and were approved by Cornell University's Institutional Animal Care and Use Committee.

## Transgenic lines and mutants

The transgenic lines, TgBAC(slc6a5:GFP) (*McLean et al., 2007*) TgBAC(slc17a6b:loxP-DsRed-loxP-GFP) (*Koyama et al., 2011*) in the *nacre* background ([*Lister et al., 1999*] *mitfa*^b692/b692) were used. Tg(glyt2:GFP) in *relaxed* background ([*Ono et al., 2001*] *cacnb1*^ts25/ts25) was used in some electrophysiology experiments (*Koyama et al., 2011*). Tg(elavl3:H2B-GCaMP6s) in the *casper* background (*Vladimirov et al., 2014*) was used in some control ablation experiments.

## Electrophysiology

Patch-clamp recordings were performed as described previously (*Koyama et al., 2011*). Tg(glyt2:GFP) was used to target glycinergic neurons. In initial experiments larvae at 4 dpf, in a *relaxed* background (*Ono et al., 2001*), in which a mutation prevents the release of calcium in internal stores in muscle, were used to immobilize the fish without a cholinergic blocker. The Tg(glyt2:GFP) line in a *nacre* background at 6 dpf was used with α-bungarotoxin paralysis (*Koyama et al., 2011*; *Lister et al., 1999*) in later experiments to confirm the physiological observations in the cancb1 background also apply to the lines used for the behavioral experiments. Feedforward glycinergic cells and Mauthner cells were targeted based on previous anatomical and physiological characterization (*Koyama et al., 2011*). Whole-cell current clamp recordings were made with a Multiclamp 700A and 700B (Molecular Devices, Sunnyvale CA) at a gain of 50 with a 500 MOhm feedback resistor (50 MOhm for the Mauthner cell) filtered at 10 kHz and digitized at 50 kHz with a Digidata 1440A (Molecular Devices) using Clampex 10.2 (Molecular Devices). The reversal potential of IPSPs was checked by injecting a series of depolarizing and hyperpolarizing currents in the postsynaptic cell, with 5 traces for each current amplitude. A glycinergic blocker, strychnine (1 μM; Sigma-Aldrich), was applied to the bath to confirm the neurotransmitter responsible for the IPSP in some experiments. Data were analyzed with MATLAB (MathWorks). The peak of each IPSP was detected and the mean was calculated for each postsynaptic cell. The morphology of the recorded cells was visualized under a confocal microscope (LSM510 META, Zeiss) after filling them with dye (0.02%, Alexa Fluor 568 or 647 hydrazide, Invitrogen) dissolved in intracellular solution. Z stacks were volume rendered using the program Imaris (Bitplane).

We made certain in triple recordings of an inhibitory neuron and two M-cells that the recordings of the cells were of high quality, with low access resistance, healthy resting potentials in the M-cells (approximately −68 mV without correction for junction potential [12 mV]), and resting potentials that were well matched between the two M-cells. Because the reversal potential for the inhibitory synapses is near resting potential, we measured the synaptic responses while injecting 1 nA of positive or negative current into the M-cells on different trials.

Putative presynaptic terminals of feedforward neurons were identified based on their characteristic puncta-like structure, and the number of puncta apposed to the cell body of Mauthner cells on each side was counted and compared with a paired Wilcoxon signed rank test with ties using the 'coin' package in R (http://cran.r-project.org/web/packages/coin/index.html).

To statistically examine whether the inhibitory strength of feedforward neurons differs between the Mauthner cells on each side, the data set for the mean peak of the IPSPs was fitted with the following generalized linear mixed model with a random effect for cell using 'lme4' package in R (http://cran.r-project.org/web/packages/lme4/index.html). The log link function for the Gaussian distribution was used to satisfy the assumptions of the linear model (normality, homogeneity of variance, independence and linearity).

$$ISPS \sim Side + Age + (1 \mid CellID)$$

The significance of the effects was examined by a likelihood ratio test of nested models.

## Femtosecond laser ablation

Tg(glyt2:EGFP) in the *nacre* background at 4 dpf were anesthetized and embedded in 1.7% low melting-point agar dissolved in 10% Hanks solution. Twenty percent Texas Red Dextran, 10,000 MW, anionic, lysine fixable (Invitrogen) was electroporated into the Mauthner cell using the protocol described previously (*Bhatt et al., 2004*). Fish were removed from the agar and immediately transferred to a small petri dish of 10% Hank's buffer after the electroporation. Fish were re-embedded again the following day (5 dpf) and imaged with a custom-made two-photon microscope (*Farrar et al., 2011*) controlled by MPScope or ScanImage (*Nguyen et al., 2011*; *Pologruto et al., 2003*). A z stack of images was acquired before the ablation using a 20× water-immersion objective lens (numerical aperture (NA)=1.0; Zeiss). 920-nm, 87-MHz, 100-fs pulses from a Ti:sapphire laser oscillator (MIRA HP; Coherent) and 1043-nm wavelength, 1-MHz, 300-fs pulses from a fiber laser (FCPA μJewel D-400; IMRA) were used to excite GFP (inhibitory cells) and Texas Red dextran (Mauthner cells). We used emission filters at 645/65 nm (center wavelength/bandwidth) and 517/65 nm (Chroma Technology) to isolate fluorescence from Texas Red dextran and GFP, respectively.

In order to ablate a cell body or a dendrite, we delivered one or two 100-fs pulses from a regenerative amplifier (800 nm wavelength, 1 kHz repetition rate; Legend, Coherent) for each target while continuously imaging the target and the surrounding region. Because the damage was mediated by an electron-ion plasma formed by nonlinear optical absorption and there was very little thermal energy deposited, the damage was largely confined to the focal volume (*Nishimura et al., 2006*). In order to ablate feedforward glycinergic neurons, we used 50 to 55 nJ pulses to target four to eight GFP positive glycinergic neurons on one side of the hindbrain located at the ventral-most part of the lateral glycinergic stripe, close to the lateral dendrite of the Mauthner cell. These were previously shown to be feedforward glycinergic neurons by retrograde labeling and paired recordings (*Koyama et al., 2011*), as confirmed by many additional recordings in this paper. In control experiments, we targeted a similar number of glycinergic neurons not implicated in escapes that were near the M-cell, but medial to the feedforward cells. In other experiments, we targeted the lateral dendrite of the Mauthner cell on one side of the hindbrain at a site next to the feedforward glycinergic neurons to examine the effects of direct damage to the M-cell. We used energy ranging from 70–90 nJ, which either successfully cut off the lateral dendrite or killed the cell, probably because the membrane failed to reseal in some cases. In each round of the experiments, we used approximately half of the fish for the ablation and the other half for controls, in which we deposited smaller energy pulses that would not damage the target (0–20 nJ for glycinergic neuron ablation and 30 – 50 nJ for lateral dendrite ablation). A z stack was acquired immediately after the ablations to examine the lesions. Fish with successful lesions were then transferred from agar into a small petri dish containing 10% Hank's buffer and allowed to recover overnight before the subsequent behavioral assay.

In order to demonstrate the spatial specificity of our ablation protocol, we used a transgenic line that expresses a genetic calcium indicator pan-neuronally to assess the damage in the cells near the target cell based on their calcium level (*Lister et al., 1999*). Feedforward glycinergic neurons were backfilled with 20 percent Texas Red Dextran, 10,000 MW, anionic, lysine fixable (Invitrogen) from the ipsilateral Mauthner cell using the procedure described previously (*Kinkhabwala et al., 2011*) in Tg(elavl3:H2B-GCaMP6s) at 3 dpf. Then the backfilled feedforward glycinergic neurons were ablated using the same protocol used above at 5 dpf while monitoring the signals from Texas Red Dextran and H2B-GCaMP6s in the target cell. Z stacks were acquired before and after the ablation.

## Behavioral assay

Escape behavior of individual larvae was examined 1 day after the ablation (6 dpf) in a petri dish (3.5 cm) filled with 10% Hank's buffer to a depth of 3–4 mm. We delivered two types of stimuli described below that elicit an escape response and filmed the fish's behavior at 1000 Hz with a high-speed video system (FASTCAM PCI, Photron). To rule out any bias in the experimental procedure and subsequent analysis, fish were coded by an individual other than the experimenter to blind the experimenter from the ablation outcomes until the escape responses were analyzed. Neuronal ablations were re-confirmed the following day (7 dpf) either with the 2-photon microscope or with a confocal microscope (LSM510 Meta, Zeiss).

In one series of experiments, escape responses to a unidirectional stimulus were examined by delivering water pulses to the tail as described previously (*Liu and Fetcho, 1999*). A pulse of colored

water (0.05 mg/ml, fast green) was delivered to the caudal body/tail by a picospritzer (~17 psi, 5 ms) through a syringe and 27 gauge needle cut blunt and bent to about 100 degrees. These settings were determined to be the lowest level that can reliably produce an escape response without the water stream disturbing the animal's movements. Stimuli were delivered to the right and left sides of the tail alternately with a minimum inter-trial interval of 2 min. At least 10 trials were collected per fish (5 trials for each side). The escape latency was calculated for each trial as the time from the contact of the pulse of water (visible because of the green dye) with the fish to the beginning of the bending movement. The trials in which fish did not show the characteristic kinematics of the escape movement (*Liu and Fetcho, 1999*) were excluded from analysis. To examine the effects of ablations on the escape latency, fish were categorized into the following groups based on the post-hoc imaging: 1) lateral feedforward inhibitory cells ablated; 2) Medial inhibitory cells ablated; 3) Mauthner's lateral dendrite cut; 4) Mauthner cell ablated; and 5) control. For statistical testing, the data for escape latency were fitted with the following linear mixed model with a random effect for subject using the 'nlme' package in R (http://cran.r-project.org/web/packages/nlme/index.html). Latency was log-transformed to satisfy the assumptions of the linear model.

$$\log(\text{Latency}) \sim \text{Ablation} * \text{Side} + (1 \mid \text{SubjectID})$$

The effects of ablations were tested by comparing the ablated and un-ablated side for each ablation group with correction for multiple comparisons using the 'multcomp' package in R (http://cran.r-project.org/web/packages/multcomp/index.html).

In another series of experiments, escape responses to an omnidirectional stimulus were examined by shaking the petri dish with a vibrating transducer (Taparia Magnetics, Mumbai, India) placed underneath it. The vibration was produced by a single 100–150 ms voltage pulse applied to the transducer, which produced a peak acceleration at pulse onset of 9 g, as measured in the vertical direction with a calibrated accelerometer (ACC-103 Omega engineering, Stamford CT). A relatively long duration pulse (as compared to the 10–20 ms duration of the initial escape bend) was used to delay any potential response to the vibration caused by the offset of the pulse. A minimum of 10 trials was collected per fish with an inter-trial interval of at least 2 min. The stimulus strength elicited escape responses reliably without fatigue at this inter-trial interval. The data were analyzed statistically by fitting the number of left versus right escapes to the following generalized linear mixed model with a random effect for subject, using a logit link function for a binomial distribution and the 'lme4' package in R (http://cran.r-project.org/web/packages/nlme/index.html).

$$\text{Direction} \sim \text{Ablation} + (1 \mid \text{SubjectID})$$

The effects of ablations were examined with Dunnett's multiple comparisons with a control by using the 'multcomp' package in R (http://cran.r-project.org/web/packages/multcomp/index.html).

## Modeling

Conductance-based models of the Mauthner circuit were built using the Brian spiking neural network simulator (http://briansimulator.org/) (*Goodman and Brette, 2008*). Basic membrane properties of the Mauthner cell and feedforward inhibitory neurons were derived from the dataset from 4 dpf fish acquired in this study (Mauthner cell: n=24; feedforward glycinergic neurons: n=28) (*Table 1*; corrected for liquid junction potential of 12 mV). Input resistance was calculated based on the current amplitude required to hyperpolarize the cell to 10 mV below the resting membrane potential. The relaxing phase of voltage traces after hyperpolarizing current injections were fitted with double exponentials using *fmincon* in MATLAB's optimization toolbox and the slowest time constant was used to calculate the capacitance of the cell (*White and Hooper, 2013*). Putative spiking threshold was estimated from phase plots (dVm/dt vs Vm) of intracellular voltage (Vm) (*Bean, 2007*). The reversal potential for chloride in the Mauthner cell was estimated from the linear fit of the IPSP amplitude and the holding potential of Mauthner cell.

Based on these parameters, the Mauthner cell and feedforward glycinergic neurons were modeled as conductance-based leaky integrate and fire neurons (Supp. Table). To represent the spike-like waveform from gap junctional inputs from the VIIIth nerve, we modeled VIII ganglion neurons using a Hodgkin-Huxley model with adjustments in the reversal potentials to set the resting membrane potentials identical to that of the Mauthner cell. The membrane area of the VIII ganglion

neurons was varied so that the number of recruited neurons increased in response to stronger stimulation. We connected one VIII ganglion neuron to only one feedforward neuron and treated them as one functional unit that recruited systematically as stimulus strength increased (*Figure 2A1–3*). The connection between them was specified as a purely electrical connection based on previous literature (*Zottoli and Faber, 1980*) and its strength was set just enough to initiate a spike in feedforward neurons in the absence of inhibitory inputs to account for the presence of feedforward inhibition even at a low stimulus strength (*Oda et al., 1995*). The relative conductance of chemical and electrical connections between the VIIIth nerve and Mauthner cell was set heuristically based on the experimentally-derived VIII nerve response in the Mauthner cell in 5 dpf zebrafish (*Figure 2A2*). We used the membrane time constant of 0.4 ms to model the eighth nerve response in the lateral dendrite based on previous literature (*Pereda et al., 2011*). The inhibitory connection from a feedforward neuron to the ipsilateral Mauthner cell was tuned based on the dataset from this study (*Figure 2A3*). The relative balance of the excitatory and ipsilateral inhibitory connections was determined so that the latency range of the Mauthner spike in response to various strengths of unilateral activation of VIII ganglion neurons matched that of the Mauthner-mediated escape to unilateral stimuli (*Liu and Fetcho, 1999*). The relative strength of contralateral and ipsilateral inhibitory inputs from feedforward neurons was determined based on the experimentally derived ratio (*Figure 1*). The strength of the putative mutual inhibition between feedforward neurons was set to recover the Mauthner spikes in response to equally strong left and right stimuli. The ablation of feedforward neurons was modeled by removing feedforward neurons on one side and their associated connections.

## Acknowledgements

We thank members of the Fetcho and Schaffer laboratories for help with experiments and comments on the work as well as Jing Yang from the Cornell Statistical Consulting Unit. We thank the Ahrens lab for sharing the H2B-GCaMP6s line. Supported by NIH grants RO1 NS 26539 and DP OD006411 to JRF and NSF CBET 1050134 to CBS.

## Additional information

### Funding

| Funder | Grant reference number | Author |
| --- | --- | --- |
| National Institute of Neurological Disorders and Stroke | RO1NS26539 | Joseph R Fetcho |
| NIH Office of the Director | DP OD006411 | Joseph R Fetcho |
| National Science Foundation | CBET1050134 | Chris B Schaffer |

The funders had no role in study design, data collection and interpretation, or the decision to submit the work for publication.

### Author contributions

MK, CBS, JRF, Designed the study, Performed or assisted with experiments and data analysis, Conception and design, Acquisition of data, Wrote the paper; FM, JS, NN, Performed or assisted with experiments and data analysis, Acquisition of data, Analysis and interpretation of data

### Author ORCIDs

Nozomi Nishimura, http://orcid.org/0000-0003-4342-9416
Joseph R Fetcho, http://orcid.org/0000-0002-3219-1169

### Ethics

Animal experimentation: All procedures conform to the US National Institute of Health guidelines regarding animal experimentation and were approved by Cornell University's Institutional Animal Care and Use Committee Protocol #2009-0084.

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
