## [Decision Letter]

Thank you for submitting your article "To turn left or right? A circuit motif in the zebrafish hindbrain for a two alternative behavioral choice" for consideration by *eLife*. Your article has been reviewed by three peer reviewers, and the evaluation has been overseen by a Reviewing Editor and a Senior Editor. The following individuals involved in the review of your submission have agreed to reveal their identity: Ronald L Calabrese (Reviewer #1 and Reviewing Editor), Michael Orger (Reviewer #2), and Stephen Soffe (Reviewer #3).

The reviewers have discussed the reviews with one another and the Reviewing Editor has drafted this decision to help you prepare a revised submission.

Summary:

In this elegant study, the authors use anatomical, electrophysiological and cell-ablation techniques to analyze how left-right escape decisions are made by the well-characterized Mauthner neuron network of zebrafish larva, an important vertebrate model genetic system. They perform impressive circuit analysis using paired and triple whole-cell recordings coupled with dye fills. They quantify synaptic interactions and use the connectivity and strength data to construct a simple model that points out the salient functional features of the feedforward (FF) inhibitory network they reveal. They then test and confirm specific predictions of the model with elegant laser ablation studies at the cellular level using simple but powerful behavioral assays. They show clearly that particular cells are involved in the Mauthner cell FF inhibitory network. These key inhibitory neurons in the circuit lie in a column of morphologically similar cells that is one of a series of such columns that form a developmental and functional ground plan for building hindbrain networks. They then argue that such FF inhibitory microcircuits may be repeated within a column for multiple left-right behavioral choices.

The writing is very clear and the figures contain the necessary data, but the figures in the pdf provided have very poor quality and contrast for most of the electrophysiological traces. Statistical analysis is appropriate. The elegance of the analysis and the general importance of the findings for the organization of inhibitory brainstem pathways and the behavioral relevance will make this paper widely interesting to the readers of *eLife*.

Essential revisions:

1) The authors should provide a clear, explicit description of what they mean by the 'motif' they have studied. It emerges from the text and is illustrated in Figure 11, but it would be helpful to be specific as to what exactly this motif does and does not comprise. A clear definition of the motif is crucial because the conclusion from the paper is that it may be fundamental to other left-right decision making processes.

2) The broad conclusion that the identified circuit motif is involved in other left-right decision making processes, while intriguing, is not completely convincing. The authors have convincingly shown the key role for the particular bilateral organization of inhibition in the M-cell escape pathway. But the M-cell pathway seems a particularly specialized pathway providing a very rapid response in which the decision about which way to turn is correspondingly fast. Many decision making processes are much slower than this, often surprisingly slow, and the motif described is not necessarily going to underlie these slower decision making processes. The authors should scale back somewhat on this conclusion or provide more convincing arguments for this idea.

3) The ability to confidently distinguish between the feedforward inhibitory neurons and other glycinergic neurons is critical for the ablation experiments. In the main text this is rather glossed over. There is more coverage in the Materials and methods section but authors should make the distinction more explicitly in the main text.

4) The model was not fully exploited and authors might consider adding a few more modeling experiments to compare with their experimental results. Specifically, consider adding a more systematic exploration of the effect of changing different aspects of connectivity in their model, e.g. symmetric vs. biased inhibition to the Mauthner cell. It would also be nice to model the effects of ablations on the output of other connectivity patterns (e.g. with/without reciprocal inhibition).

---

## [Author Response]

Essential revisions:

*1) The authors should provide a clear, explicit description of what they mean by the 'motif' they have studied. It emerges from the text and is illustrated in Figure 11, but it would be helpful to be specific as to what exactly this motif does and does not comprise. A clear definition of the motif is crucial because the conclusion from the paper is that it may be fundamental to other left-right decision making processes.*

We have added to the Discussion a section (third paragraph) that explicitly describes what we see as the key features of the motif. We hope that this, along with the figure showing it, will make this unambiguous to the reader.

*2) The broad conclusion that the identified circuit motif is involved in other left-right decision making processes, while intriguing, is not completely convincing. The authors have convincingly shown the key role for the particular bilateral organization of inhibition in the M-cell escape pathway. But the M-cell pathway seems a particularly specialized pathway providing a very rapid response in which the decision about which way to turn is correspondingly fast. Many decision making processes are much slower than this, often surprisingly slow, and the motif described is not necessarily going to underlie these slower decision making processes. The authors should scale back somewhat on this conclusion or provide more convincing arguments for this idea.*

We did not actually conclude, but only suggested/predicted, that a similar motif might be involved in other such choices. This is important, as we tried to be very careful to separate conclusions based on our data from their potential broader implications. We used the wording in the Discussion: “might expect”, “raises the possibility”, “we therefore predict”. We do not disagree that the M-cell circuit is specialized for a rapid choice. But, the use of repetitive arrays of a cell type that operate over a range of time scales, as exists in hindbrain based on our earlier work, makes it hard for us not to raise the possibility that “slower” choices will be made using different (younger and slower circuits) with the same overall cell types and circuit topology. Many responses in larval fish are fast (less than 100 millisecond time frame), so they may include pathways as direct as those in the M-cell circuit that have neurons with different cellular properties, and/or populations of cells that are recruited more slowly by sensory stimuli to make choices over longer time frames. We have now included a more explicit statement of this in the last paragraph of the Discussion.

We hope that the editors and reviewers will allow us to present our testable ideas that could be of large consequence. We too often focus on what makes each circuit special, but developmental and functional topographies in hindbrain and spinal cord suggest that many circuits arise from the same neuronal ground plan, so we think it is just as important to ask what different circuits (even ones operating over different time scales) share if we are to arrive at broader principles of circuit construction.

*3) The ability to confidently distinguish between the feedforward inhibitory neurons and other glycinergic neurons is critical for the ablation experiments. In the main text this is rather glossed over. There is more coverage in the Materials and methods section but authors should make the distinction more explicitly in the main text.*

We have now made this clear by including more information about the FF cell location and the more medial control ablated cells in the text proper. As we indicate in the text, both lie adjacent to the M-cell, but in a column located about 40 microns medial to the one with the FF cells (subsection “Testing of behavioral predictions”, fifth paragraph).

*4) The model was not fully exploited and authors might consider adding a few more modeling experiments to compare with their experimental results. Specifically, consider adding a more systematic exploration of the effect of changing different aspects of connectivity in their model, e.g. symmetric vs. biased inhibition to the Mauthner cell. It would also be nice to model the effects of ablations on the output of other connectivity patterns (e.g. with/without reciprocal inhibition).*

We have added the suggested modeling including a comparison of symmetric versus biased inhibition and an exploration of the ablation effects with and without the reciprocal inhibition (Figure 2—figure supplement 1). We also modeled removal of just half of the inhibitory neurons, rather than all of them as in the original version. This is more akin to our actual perturbation and still leads to the prediction of the shorter latency we observed. We modeled ablation in a case without the mutual inhibition, as the reviewer suggested, but it was minimally informative because it retained a large area of no responses like the models without ablation, so we do not include it here. We have, however, incorporated most of the new modeling into the paper.